# Candida Administration in Bilateral Nephrectomy Mice Elevates Serum (1→3)-β-D-glucan That Enhances Systemic Inflammation Through Energy Augmentation in Macrophages

**DOI:** 10.3390/ijms22095031

**Published:** 2021-05-10

**Authors:** Jiraphorn Issara-Amphorn, Cong Phi Dang, Wilasinee Saisorn, Kavee Limbutara, Asada Leelahavanichkul

**Affiliations:** 1Translational Research in Inflammation and Immunology Research Unit (TRIRU), Department of Microbiology, Chulalongkorn University, Bangkok 10330, Thailand; jiraphorn298@gmail.com; 2Department of Microbiology, Faculty of Medicine, Chulalongkorn University, Bangkok 10330, Thailand; pilotdang1308@gmail.com (C.P.D.); wsaisorn@gmail.com (W.S.); 3The Center of Excellence in Systems Biology, Faculty of Medicine, Chulalongkorn University, Bangkok 10330, Thailand; xcube.md@gmail.com; 4Immunology Unit, Department of Microbiology, Chulalongkorn University, Bangkok 10330, Thailand

**Keywords:** uremia mice, bilateral nephrectomy, Candida, endotoxin, (1→3)-β-D-glucan

## Abstract

Systemic inflammation, from gut translocation of organismal molecules, might worsen uremic complications in acute kidney injury (AKI). The monitoring of gut permeability integrity and/or organismal molecules in AKI might be clinically beneficial. Due to the less prominence of Candida albicans in human intestine compared with mouse gut, C. albicans were orally administered in bilateral nephrectomy (BiN) mice. Gut dysbiosis, using microbiome analysis, and gut permeability defect (gut leakage), which was determined by fluorescein isothiocyanate-dextran and intestinal tight-junction immunofluorescent staining, in mice with BiN-Candida was more severe than BiN without Candida. Additionally, profound gut leakage in BiN-Candida also resulted in gut translocation of lipopolysaccharide (LPS) and (1→3)-β-D-glucan (BG), the organismal components from gut contents, that induced more severe systemic inflammation than BiN without Candida. The co-presentation of LPS and BG in mouse serum enhanced inflammatory responses. As such, LPS with Whole Glucan Particle (WGP, a representative BG) induced more severe macrophage responses than LPS alone as determined by supernatant cytokines and gene expression of downstream signals (*NFκB*, *Malt**-1* and *Syk*). Meanwhile, WGP alone did not induced the responses. In parallel, WGP (with or without LPS), but not LPS alone, accelerated macrophage ATP production (extracellular flux analysis) through the upregulation of genes in mitochondria and glycolysis pathway (using RNA sequencing analysis), without the induction of cell activities. These data indicated a WGP pre-conditioning effect on cell energy augmentation. In conclusion, Candida in BiN mice accelerated gut translocation of BG that augmented cell energy status and enhanced pro-inflammatory macrophage responses. Hence, gut fungi and BG were associated with the enhanced systemic inflammation in acute uremia.

## 1. Introduction

Acute kidney injury (AKI) is an important cause of death in critically ill patients [1]. AKI results in an accumulation of uremic toxins that induce systemic inflammation [2]. Likewise, the uremia-induced inflammation worsens infections and sepsis [3,4] as the severe sepsis in a two-hit injury of sepsis on top of uremia is mentioned [4]. Moreover, accumulation of uremic toxins results in multisystem dysfunctions as a crosstalk between the kidney and the other remote organs in both directions is mentioned. Accordingly, the gut–liver–kidney axis is an example of the important association among organs during uremia [5]. On the other hand, gut permeability barrier consists of only a single layer of the gut epithelium with an approximate surface area at 32 m^2^ that separates between the host and the organisms in gut contents [6]. As such, gut permeability defect (gut leakage) leads to the translocation of several organismal molecules from the gut into blood circulation [7,8,9,10,11]. Subsequently, gut translocation of these organismal molecules, referred to as pathogen associated molecular patterns (PAMPs), profoundly induces systemic inflammation [5]. In AKI, limited excretion of uremic toxins through kidneys leads to an enhanced toxin elimination through the gastrointestinal tract that results in a selective growth of some gut bacteria (gut dysbiosis) [5,12]. Subsequently, gut dysbiosis damages the intestinal mucosa that causes gut translocation of lipopolysaccharide (LPS), the most abundant PAMPs in the gut (referred to as “endotoxin”) [5,12]. Then, AKI worsens gut dysbiosis and the dysbiosis deteriorates AKI. Indeed, the attenuation of gut leakage with several strategies improves uremic complications in patients [13]. Despite the high abundance of LPS in gut contents due to the prominence of Gram-negative bacteria in feces, gut fungi (especially Candida albicans) are the second most predominant gut organisms in humans [14] and the influence of gut fungi in clinical practice is well-known [15]. As such, the increased intestinal Candida colonization in some patient conditions (alcoholic liver cirrhosis [16,17,18] or broad-spectrum antibiotic use [14]) is also mentioned. In addition, the presence of (1→3)-β-D-glucan (BG), a major component of Candida cell wall, in serum (from gut translocation) worsens the endotoxemia-induced inflammation [19,20], partly through the synergy between Toll-like receptor 4 (TLR-4) and Dectin-1, the receptors of LPS and BG, respectively [21]. Because of the lesser abundance of C. albicans in mouse gut than human intestine, the presence of intestinal C. albicans in mouse models might resemble more to that of human conditions [22].

While Candida spp. in patients is detectable by stool culture [23], the abundance of Candida in mouse feces is not high enough to be detectable by culture method [22]. Although gut translocation of the viable fungi is more difficult than the intact bacteria due to the larger size of fungi [1], the similar sizes of LPS and BG, the main cell wall components of bacteria and fungi, respectively, result in the elevation of both molecules in serum during gut leakage [23]. Moreover, gut leakage is also directly inducible by uremia-induced enterocyte damage [24] and the presence of gut fungi worsens gut leakage through either the direct intestinal invasion or the indirect fungi-induced bacterial dysbiosis [5]. On the other hand, macrophages are the important innate immune cells that are responsible for the initiation phase of inflammation [25]. Macrophages also distribute throughout the body, referred to as “sentinel immune cells” in several organs, including the liver (Kupffer cells), the brain (microglia), the kidney (mesangial cells), the lung (alveolar macrophages), and the intestine (intestinal macrophages) [26,27,28]. Additionally, macrophages are the heterogeneous immune cells with pleiotropic functions; the pro- and anti-inflammatory properties [29,30,31], which use glycolysis and mitochondria, respectively, as the main metabolic signatures [32,33,34]. Interestingly, the cell energy status of macrophages could be altered by either PAMPs, including endotoxin [35], or damage associated molecular patterns (DAMPs), including uremic toxins [36]. Although the exacerbation of macrophage inflammatory responses after activation by uremic toxins and endotoxin in synergy [36] through the cross-talk between TLR-4 and Dectin-1 [37,38,39] is previously demonstrated, an impact of fungal molecules on macrophage energy status is still interesting. Furthermore, BG also accelerates macrophage responses against endotoxin through induction of the trained innate immunity, an ability of innate immune cells to prolong the hyperactive state after the first stimulation by some specific agents and respond nonspecifically to the subsequent activations [40]. Because (i) there are still only a few studies on the topic of gut fungi in uremia and (ii) the presence of Candida in mouse gut might more resemble patient conditions, the uremic mouse model with Candida administration is an interesting tool for the exploration on impact of fungi in the model. Hence, we performed an acute uremia model in mice with C. albicans pretreatment before the bilateral nephrectomy (BiN) surgery (BiN-Candida) to examine the impact of C. albicans on acute uremia.

## 2. Results

Gut Candida enhanced inflammatory responses at 48 h post-surgery through a gut permeability defect that elevated LPS and BG in serum (gut translocation). The macrophage energy augmentation by BG was, at least in part, responsible for the synergistic effect of LPS and BG in acute uremia.

### 2.1. Candida Administration Worsened Uremia-Induced Systemic Inflammation through the Induction of Gut Dysbiosis and Gut Translocation of LPS and BG in Bilateral Nephrectomy Mice

At 48 h post-surgery, C. albicans administration increased the fungal burden in feces and elevated systemic inflammation (serum cytokines), but did not alter serum creatinine (Figure 1A–E). Candida in BiN damaged gut permeability integrity as determined by FITC-dextran assay, the elevation of LPS and BG in serum and the reduction in enterocyte tight junction molecules (Claudin-1 and Occludin) (Figure 1F–K). Despite a similar level of serum creatinine, a current renal injury biomarker (Figure 1B), gut Candida prominently reduced bacterial diversity (an indicator of the intestinal healthiness) in BiN with Candida (BiN-Candida) than non-Candida BiN as indicated by total operational taxonomy unit (OTUs), Chao1 richness estimation score (a variety of the bacterial species in the population) and Shannon evenness determination score (the similarity of bacterial abundance among all bacterial species in the population) (Figure 2E). However, BiN-Candida demonstrated a lesser abundance of Firmicutes, a group of bacteria with possible beneficial properties [41], and Proteobacteria, a group of pathogenic bacteria [42] (Figure 2F). Hence, the more severe fecal dysbiosis in BiN-Candida than non-Candida BiN might induce the higher translocation of LPS and BG from the gut into blood circulation (gut leakage).

In macrophages, LPS (without BG) induced inflammatory responses as indicated by supernatant cytokines (TNF-α, IL-6, and IL-10) and gene expression of downstream signals (*NFκB* and *Malt**-1*, but not *Syk*) (Figure 3A–E). The activation by BG (without LPS), using Whole Glucan Particle (WGP) as a representative BG, did not activate macrophages (Figure 3A–E). However, LPS with BG (LPS+WGP) induced the more prominent macrophage responses than LPS alone, as indicated by supernatant cytokines (TNF-α and IL-10) and the downstream signals (*NFκB*, *Syk* and *Malt**-1*) (Figure 3A–F). Notably, *Syk* is a common downstream signaling of both TLR-4 and Dectin-1, a receptor for LPS and BG, respectively [37,38,39]. Additionally, LPS+WGP and WGP alone, but not LPS alone, reduced protein abundance of phosphorylated AMP-activated protein kinase (p-AMPK), a cell energy sensor [43] (Figure 3G,H), implying a status of the high cell energy level after WGP activation. For other macrophage functions, the activation by LPS with or without WGP did not alter macrophage phagocytic activity, while all of the activations decreased bactericidal activity as the viable bacteria in stimulated macrophages were higher than the control group (Figure 3I).

### 2.2. The Acceleration of Cell Energy in Macrophages by BG, an Impact on the Enhanced Pro-Inflammatory Responses

Because of (i) the presence of both LPS and BG in mouse serum after BiN (Figure 1G,H), (ii) the possible alteration in cell energy status by organismal molecules, especially LPS [29,30,31], and (iii) the lack of data on cell energy status after activation by LPS together with BG, the extracellular flux analysis was tested in macrophages. Indeed, the mitochondrial stress test using several mitochondrial blocking agents (Oligomycin and Rotenone/Antimycin A) and a mitochondrial augmentation substance (FCCP) (Figure 4A,B) was performed. As such, mitochondrial activity in macrophages with WGP (WGP or WGP+LPS) was higher than LPS-activated macrophages, which was not different from the control group (Figure 4C). Most of the mitochondrial parameters from macrophages with WGP alone or WGP+LPS were higher than LPS-activated macrophages, including basal respiration, maximal respiration, ATP production, proton leak, and non-mitochondria respiration (Figure 4D,E), but not mitochondrial spare capacity (data not shown). In mitochondrial parameters, only basal respiration and maximal respiration were shown (Figure 4D,E) because of the similar patterns among these parameters. With RNA-sequencing analysis, several groups of mitochondria-associated genes were highly upregulated in WGP-activated macrophages (without LPS) compared with other groups, including *SLC25A33* (a genes for imports/exports pyrimidine nucleotides into and from mitochondria), *FASTKD3* (Fas-activated serine/threonine kinase domain, a regulator for mitochondrial energy-balance), *COQ10B* (a coenzyme Q in the respiratory chain), *MTFR2* (Mitochondrial Fission Regulator 2), and *PRELID1* (protein of relevant evolutionary and lymphoid interest for the transfer of phosphatidic acid between liposomes) (Figure 4F), that might be associated with mitochondrial energy augmentation by WGP. Notably, these WGP-activated genes were upregulated in RNA sequencing analysis despite the non-inflammatory response after activation by WGP alone (Figure 3A–K). In addition, there was a similar direction of the gene expression between LPS and WGP+LPS groups, except for *MTFR2* (a gene for mitochondrial fission). As such, *MTFR2* highly upregulated in WGP+LPS activated macrophages, while downregulated in LPS activated cells (Figure 4F). These data implied a possible acceleration in mitochondrial fission by WGP to create the new mitochondria [44] that might be associated with the increased cell energy status.

Likewise, glycolysis stress test by a glycolysis augmentation procedure using a mitochondrial blocking agent (Oligomycin) and a glycolysis inhibitor (2-DG) was performed (Figure 5A,B). Interestingly, WGP enhanced glycolysis activity in macrophages (Figure 5C), especially glycolysis (a parameter after cell induction by glucose) and glycolysis capacity (an ability to increase glycolysis activity in compensation for mitochondrial blockage) (Figure 5D,E). However, Oligomycin did not enhance glycolysis reserve in macrophages (Figure 5D,E), which might be due to the low glycogen reserve of macrophages [45]. In parallel, LPS (without WGP) increased glycolysis capacity (Figure 5C–E) and cytokine production in macrophages (Figure 3A–C) supporting the glycolysis dependent macrophage cytokine production [46]. With RNA sequencing analysis, WGP (without LPS) upregulated several enzymes in glycolysis pathway in macrophages compared with control group, including *ALDOA* (Aldolase, Fructose-Bisphosphate A), *PKM* (Pyruvate kinase isozymes), *ENO1* and *ENO2* (Enolase), *GAPDH* (Glyceraldehyde 3-phosphate dehydrogenase), *PGK1* (Phosphoglycerate kinase), and *LDHA* (lactate dehydrogenase A, an enzyme for a conversion of pyruvate into lactate) (Figure 5F). Meanwhile, WGP (without LPS) downregulated only a few genes in macrophages compared with the control group, including *PFKM* (Phosphofructokinase) and *PGAM2* (Phosphoglycerate mutase) (Figure 5F). On the other hand, there was a similar direction of the alteration in most genes between LPS alone versus WGP+LPS in macrophages (Figure 5F). However, *ENO2*, *PGK1*, and *LDHA* were upregulated in WGP+LPS group when compared with LPS stimulation alone (Figure 5F). These data suggested a glycolysis acceleration by WGP that possibly enhance cell energy status. Despite limited macrophage activation (Figure 3A–I), WGP activated several enzymes in both the mitochondria (Figure 4A–F) and glycolysis pathway (Figure 5A–F).

Hence, WGP accelerated both mitochondrial and glycolysis activities of macrophages (Figure 4C and Figure 5C) that might be responsible for the synergistic effect of WGP+LPS (Figure 3A–I). The extracellular flux analysis machine could identify the source of ATP (from mitochondria or glycolysis pathways) by the analysis of ATP production using mitochondrial blocking agents (Oligomycin and Rotenone/Antimycin A) (Figure 6A). As expected, ATP from mitochondria and glycolysis after activation by WGP, with or without LPS, were higher than LPS stimulation alone, which was similar to the control group (Figure 6B,C). The cell energy phenotype profile of macrophages after LPS activation (without WGP) was mildly elevated but was still in a range of the quiescent energy profile (Figure 6D). On the other hand, WGP stimulation, alone or with LPS, augmented cell energy status into the area of energetic profile (Figure 6D). In control macrophages, approximately 90% and 10% of cellular ATPs were produced from mitochondria and glycolysis, respectively (Figure 6E,F). Meanwhile, approximately 60% and 40% of macrophage ATPs were derived from mitochondria and glycolysis, respectively, after WGP activation (Figure 6E,F). In parallel, there was no difference in percentage of macrophage ATPs from mitochondria and glycolysis between the activation by LPS and WGP+LPS (Figure 6E,F). Likewise, there was a similar ATP production rate (from mitochondria and/or glycolysis) either between control macrophages and LPS-activation (without WGP) or between WGP alone and WGP+LPS activation (Figure 6G, solid lines). However, mitochondrial ATP production rate in WGP+LPS activated macrophages was higher than WGP activated cells (Figure 6G, dotted line).

With RNA sequencing analysis, WGP activation (without LPS) upregulated several enzymes of glycolysis pathways when compared with control macrophages, including *ALDOS*, *PRKAA2* (5’-AMP-activated protein kinase catalytic subunit alpha-2; also referred to as AMPK), *PKM*, *ENO* (1 and 2), *LDHA*, *PGK1*, *PFKL* (Phosphofructokinase isoform), *PGM1* (Phosphoglucomutase-1), and *HK1* (Hexokinase-1). In parallel, WGP activation (without LPS) downregulated *PFKM*, *P2RX7* (ionotropic ATP-activated receptor), and *PINK1* (PTEN-induced kinase 1, a mitochondrial serine/threonine-protein kinase) (Figure 6H). Among these ATP productions associated genes, most genes are coded for enzymes in glycolysis pathway except for *PINK**-1* that is represented for a mitochondrial enzyme [47]. Notably, direction of the gene expression in LPS-activated macrophages was similar to WGP+LPS stimulated cells except for the increase in glycolysis enzymes [48]; *LDHA*, *PGK1*, and *ENO2*, after WGP+LPS activation (Figure 6H). These data implied an impact of WGP on enhanced ATP production, perhaps mainly through the acceleration of the glycolysis pathway. Furthermore, comparisons between the biological groups of genes (DEGs) (Figure 7A–E) in macrophages with various treatments were performed.

Accordingly, WGP enhanced groups of the genes for glycolysis activity, while LPS or LPS+WGP upregulated genes for the DNA and cytokines synthesis, when compared with control macrophages (Figure 7A–C). In WGP+LPS activation, groups of the genes for inflammatory process and cytokine activation were upregulated when compared with the stimulation by WGP or LPS alone (Figure 7D,E). These data implied the pro-inflammatory potency of LPS that was accelerated by WGP [49]. Hence, WGP, a structural molecule of the fungal cell wall, accelerated energy production (ATP synthesis) in macrophages possibly through the upregulation of genes in mitochondria (*MTFR2*) and glycolysis pathway (*LDHA*, *PGK1*, and *ENO*) that might be responsible for the synergy of WGP and LPS. The utilization of WGP to enhance macrophage functions and the blockage of these enzymes to reduce macrophage pro-inflammation is interesting for several diseases.

## 3. Discussion

Gut fungi in acute uremia enhanced systemic inflammation through gut dysbiosis-induced translocation of organismal molecules and the macrophage energy augmentation by BG.

### 3.1. Gut Dysbiosis in Candida Administered Mice with Bilateral Nephrectomy Induced Gut Permeability Defect and the Translocation of Organismal Molecules from Intestinal Contents

Uremia-induced gut dysbiosis, partly through the selective growth of some bacteria by the uremic toxins in the gut, leads to the reduction in beneficial bacteria (short-chain fatty acid and mucin producing bacteria) and the toxin-induced enterocyte damage [50,51,52,53,54,55]. Because uremia induces gut inflammation that enhances intestinal fungal colonization and systemic fungal infection [56] through the impaired mucosal immunity [57,58], the intestinal integrity and fungal abundance in the gut should be considered in acute kidney injury. However, data on the influence of uremic toxins during the presence of gut fungi is still lacking. Because C. albicans in mouse feces are detectable only by PCR, but not by culture [22,59], different from the human condition [23], C. albicans administration is necessary to induce Candida presentation in the mouse intestines [20,60,61]. Indeed, fecal Candida was not detectable at 48 h post-BiN (Figure 1A), supporting the necessity of Candida administration in several doses on the model. Indeed, systemic inflammation in BiN-Candida was more severe than non-Candida BiN, highlighting the difference between the conventional model without Candida and our current model. In parallel, gut permeability defect, referred to as “gut leakage” [5], and gut translocation of organismal molecules (LPS and BG in serum) in BiN-Candida was more severe than non-Candida BiN, supporting a previous publication [36]. However, it is uncertain that Candida worsens bacterial dysbiosis in the gut of BiN mice as the pathogenic bacteria, Bacteroides and Proteobacteria [62], in BiN-Candida mice were less prominent than non-Candida BiN mice. Likewise, Gram-negative bacteria in feces (the source of LPS in gut contents), as calculated from fecal microbiome analysis in phylum level, were not different among groups (Sham, BiN, BiN-Candida), despite the increase in BG in gut contents of BiN-Candida mice. Hence, the higher serum BG in BiN-Candida mice than non-Candida BiN mice is a result of (i) the increased BG in gut contents and (ii) the enhanced gut leakage through the direct enterocyte damage [63,64] by Candida and the indirect bacterial dysbiosis (but not the alteration of LPS level in gut contents). Nevertheless, our data indicated the difference between the mouse models with and without gut Candida and the model with gut Candida might better resemble the human condition.

### 3.2. Prominent Macrophage Inflammatory Responses to LPS with BG, When Compared with LPS Alone, an Impact of BG-Induced Cell Energy Augmentation

Although uremia-induced serum LPS is well known [65], data on the influence of LPS with the presence of BG (LPS+BG) is still lacking. Here, WGP was used as a BG representative because of the specific activation on Dectin-1 without the stimulation on TLR-4 in this form of BG [66,67]. With LPS+BG activation, the crosstalk between TLR-4 and Dectin-1 [68] is, at least in part, responsible for the LPS-BG synergistic effect as demonstrated by profound supernatant cytokines in the activated macrophages as previously published [37,38,39]. Indeed, LPS and BG possibly activated NFκB transcriptional factor through Malt-1 and Syk [69,70] that resulted in pro-inflammatory macrophage responses. Despite the more prominent cytokine production after LPS+BG activation than LPS stimulation alone, LPS+BG did not enhance phagocytosis and bactericidal activity. These data implied a limited organismal control of macrophages with a profound pro-inflammatory cytokine production during the presence of both LPS and BG that might worsen disease severity of sepsis (the hyper-responsiveness toward severe infection). In parallel, the reduced p-AMPK (an energy sensor molecule) in LPS+BG implied an elevation of cell energy status in macrophages that might be associated with high cytokine production. As such, AMPK depletion is associated with pro-inflammatory responses and AMPK elevation maintains energy homeostasis through reduction in the pro-inflammatory state and activation of the alternative energy sources (lipid metabolism) [71,72,73,74,75]. In addition, metabolic homeostasis is associated with macrophage responses [76] as glycolysis and mitochondrial oxidative phosphorylation are major pathways for energy production during pro- and anti-inflammatory responses, respectively [30,77,78].

With extracellular flux analysis, LPS slightly increased glycolysis capacity (a capacity to increase ATP production after mitochondrial blockage), without mitochondrial activation as previously published [79]. As such, the LPS enhanced glycolysis was possibly associated with high supernatant cytokine production. On the other hand, the combination of LPS+BG (WGP) not only increased glycolysis parameters, but also enhanced most of the mitochondrial activities. Moreover, the enhanced macrophage energy status by LPS+BG was also associated with the increased cytokine production (TNF-α and IL-10). These data supported a correlation between the increased macrophage energy status (from mitochondria and glycolysis) and the cytokine production activity (but not phagocytosis and bactericidal activity). Indeed, the decreased macrophages cytokine production through the blockage on either glycolysis [30] or mitochondrial activity [75,80] are reported. In addition, cell energy is necessary for several processes of the cytokine production (referred to as ATP-dependent cytokine production [81,82]), including substrate preparation and protein excretory processes [82].

In contrast, BG (WGP) alone highly accelerated both glycolysis and mitochondrial activities in macrophages, despite a neutral effect on cytokine production. The BG-enhanced cell energy status possibly prepared macrophages for the subsequent LPS stimulation in similar to the enhanced metabolic activity in BG-primed monocytes [83]. Despite the inactive cytokine production, BG upregulated several enzymes in glycolysis and mitochondrial pathways that increased ATP synthesis (compared with control). Indeed, subsequent LPS activation in BG-primed macrophages, with the high cell energy status, induced prominent cytokine excretion. Similarly, BG-primed human monocytes demonstrate metabolic reprogramming and profoundly produce cytokines after the subsequent stimulation by other organismal molecules, referred to as “trained immunity” [40]. While the enhanced inflammatory activity is beneficial for organismal control [40], the hyper-responsiveness of BG-primed macrophages in acute uremia might cause the unnecessary inflammation that worsens uremic complications.

### 3.3. Clinical Translation and Utilization of BG for Macrophage Manipulation

Although more studies on this topic are necessary for a solid conclusion, a proof of concept in a possible impact of gut fungi on acute uremia through BG-induced metabolic reprogramming in macrophages is demonstrated (Figure 8). A proper evaluation of gut fungi and gut leakage monitoring (serum endotoxin and serum BG) might be helpful in acute uremia as several treatments for uremia-induced gut leakage, including probiotics and short-chain fatty acid, are mentioned [84]. On the other hand, the primed cell energy status by WGP is also interesting for macrophages harness to the proper direction. This might be helpful for the treatment of macrophage exhausting conditions, including tumor-associated macrophages and sepsis-induced immune suppression [85,86]. However, the ex vivo experiments on macrophages that are directly separated from BiN mice with versus without Candida administration were not performed. More studies on these topics are interesting.

## 4. Materials and Methods

### 4.1. Animals and Animal Model

The protocol of animal care and use was approved by the Institutional Animal Care and Use Committee of the Faculty of Medicine, Chulalongkorn University, Bangkok, Thailand, following the U.S. National Institutes of Health (NIH). Accordingly, male 8-week-old C57BL/6 mice from Nomura Siam International (Pathumwan, Bangkok, Thailand) were used. Then, 1 × 10^6^ CFU of Candida albicans (the American Type Culture Collection; ATCC 90028) (Fisher Scientific, Waltham, MA, USA), were cultured overnight on Sabouraud dextrose broth (SDB) (Oxoid, Hampshire, UK) at 35 °C for 48 h before enumeration by a hemocytometer. After that, C. albicans diluted in 0.5 mL phosphate buffer solution (PBS) or PBS alone was orally administered for 7 days. After that, bilateral nephrectomy (BiN) was performed 6 h after the last dose of C. albicans through abdominal incision as previously mentioned [4,7,36,87]. In the sham group, both kidneys were only identified by abdominal incision before closing. Fentanyl at 0.03 mg/kg of body weight in 0.5 mL normal saline solution (NSS) was subcutaneously administered after the operation for analgesia and fluid replacement. Then, mice were sacrificed with cardiac puncture under isoflurane anesthesia with blood and intestinal collection. Due to the most predominant of gut organisms (Gram-negative bacteria and fungi) in the colon [88] and the similarity of uremia-induced gut permeability defect between small bowel and large bowel [36], only ascending colons (distal to the caecum) were kept in tissue frozen in optimal cutting temperature (OCT) compound (Tissue-Tek OCT compound; Sakura Finetek USA, Inc., Torrance, CA, USA) for fluorescence microscopic evaluation. Additionally, feces from all parts of the colon were combined for fecal microbiome analysis and fecal fungal burden determinations (culture method).

### 4.2. Mouse Sample Analysis and Immunofluorescence

Feces were suspended with PBS (100 mg feces per 1 mL PBS), serially diluted before plating onto Sabouraud dextrose agar (SDA) (Oxoid) and incubated at 35 °C for 72 h before colony enumeration of fecal fungal burden. Serum creatinine and serum cytokines were determined by QuantiChrom creatinine assays (DICT-500, Bioassay, Hayward, CA, USA) and the enzyme-linked immunosorbent assay (ELISA) (Invitrogen, Carlsbad, CA, USA), respectively. Gut leakage was determined by (i) the detection of serum fluorescein isothiocyanate-dextran (FITC-dextran), a nonabsorbable high-molecular-weight molecule, after an oral administration, (ii) the elevation of pathogen associated molecular patterns (PAMPs) from gut contents including LPS and BG and (iii) the evaluation of intestinal tight junction molecules following a previous publication [36]. Briefly, FITC-dextran (molecular weight, 4.4 kDa) (Sigma-Aldrich, St. Louis, MO, USA) at 0.5 mL (12.5 mg) was orally administered at 3 h prior to sacrifice and blood was collected at sacrifice before analyzing by fluorescence spectroscopy (Varioskan Flash) (Thermo Scientific, Carlsbad, CA, USA) at excitation and emission wavelengths of 485 and 528 nm, respectively, with a standard curve of FITC-dextran. Serum LPS and BG were measured by HEK-Blue LPS detection (InvivoGen, San Diego, CA, USA) and Fungitell (Associates of Cape Cod, Inc., East Falmouth, MA, USA), respectively. The values of LPS and BG below 0.01 and 7.8, respectively, were recorded as 0 due to being beyond the lower limit of the standard curve.

For the immunofluorescent staining, 5-mm-thick frozen tissue sections were fixed with acetone, blocked with 1% bovine serum albumin (BSA) in 10% fetal bovine serum (FBS) and stained with primary antibody against enterocyte tight junction molecules of Claudin-1 and Occludin (Thermo Fisher Scientific, IL, USA) followed by the secondary antibody (Alexa Fluor 546 goat anti-rabbit IgG) (Life Technologies, USA). After that, the slides were stained with 4’,6-diamidino-2-phenylindole (DAPI; BioLegend, USA) for nuclei staining, mounted (Prolong; Life Technologies), visualized, and scored (percentage of fluorescent area) using a Zeiss LSM 800 confocal microscope (Carl Zeiss, USA).

### 4.3. Fecal Microbiome Analysis

Fecal microbiome analysis was performed according to the previous publications [20,36,89]. Briefly, feces from individual mouse at 0.25 g were used for total DNA extraction by a power DNA isolation kit (MoBio, Carlsbad, CA, USA) and determined metagenomic DNA quality by agarose gel electrophoresis with nanodrop spectrophotometer. Universal prokaryotic forward primer 515F (5’-GTGCCAGCMGCCGCGGTAA-3’) and reverse primer 806R (5’-GGACTACHVGGGTWTCTAAT-3’), with appended 50 Illumina adapter and 30 Golay barcode sequences, were used for 16S rRNA gene V4 library construction. Triplicate PCRs were performed. For 16S rRNA purification, GenepHlow gel extraction kit (Geneaid Biotech Ltd., New Taipei City, Taiwan) was used and the quantification was performed using PicoGreen (Invitrogen, Eugene, OR, USA). MiSeq300 sequencing platform (Illumina, San Diego, CA, USA) was applied with previously described index sequences [90]. Mothur’s standard quality screening operating procedures for the MiSeq platform with aligned and assigned taxa (operational taxonomic units [OTUs]) based on default parameters were also used [91].

### 4.4. Macrophage Cytokines and Western Blot Analysis

Macrophages were derived from bone marrow following a published protocol [92]. Briefly, mouse bone marrow was flushed from the femurs and tibias of 8-week-old mice, plated in petri dishes in modified Dulbecco’s Modified Eagle Medium (DMEM). After that, macrophage colony-stimulating factor (M-CSF) (Sigma-Aldrich) were added at 4 days and removed from the petri dishes at 7 days of the procedure. Then, macrophages (1 × 10^5^ cells/well) were incubated with a representative of (1→3)-β-D-glucan (BG) using whole glucan particle (WGP), the purified BG from Saccharomyces cerevisiae (WGP Dispersible; Biothera, Eagan, MN, USA). As such, WGP at 500 µg/mL (WGP) with or without LPS (Escherichia coli 026: B6; Sigma–Aldrich) at 100 ng/mL or the control (DMEM alone) were incubated for 6 h before supernatant cytokines measurement (PeproTech, Oldwick, NJ, USA). The cells were harvested by cold PBS before centrifugation at 2000 rpm at 4 °C for 5 min and were stored at −80 °C until used. Additionally, the western blot analysis followed a previous protocol was performed [35]. Briefly, 20 µg of the homogenized macrophages as determined by Bicinchoninic acid assay (BCA) (Thermo Fisher Scientific) and was used for Sodium Dodecyl Sulfate polyacrylamide gel electrophoresis (SDS-PAGE). Subsequently, the proteins were transferred to Polyvinylidene fluoride (PVDF) membrane before incubation with specific primary antibodies against 5’ adenosine monophosphate-activated protein kinase (phosphorylated-AMPK or p-AMPK) (Cell signaling, Beverly, MA, USA) or Beta-actin (β-actin) (Cell signaling) overnight at 4 °C. Then, a secondary antibody linked with horseradish peroxidase enzyme was incubated for 1 h before visualization by ImageQuantTM LAS 500 (GE-Healthcare, Little Chalfont, Buckinghamshire, UK).

### 4.5. Real-Time Polymerase Chain Reaction and RNA Sequencing Analysis

The expression of several genes was identified using real time polymerase chain reaction (PCR). As such, total RNA was extracted using an RNeasy mini kit (Qiagen, Hilden, Germany) and the cDNA synthesis (0.3 µg of total RNA) was performed using a high capacity reverse transcription assay (Applied Biosystems, Warrington, UK) according to the manufacturer’s instructions. Real-time PCR was performed using an Applied QuantistudioTM 7 Flex Real-Time PCR System (Thermo Scientific) with SYBR® Green PCR Master Mix (Applied Biosystems). The results were interpreted in terms of relative quantitation using the comparative threshold (delta-delta Ct) method (2^−ΔΔCt^). The expression of target genes, using several primers (Table 1), in the samples, normalized to β-actin (an endogenous housekeeping gene) was demonstrated.

The RNA sequencing analysis was performed to determine the influence of WGP and LPS, alone or in combination, to macrophages. Briefly, macrophages after the 6 h activation by WGP (Biothera) (500 µg/mL) with or without LPS (Sigma Aldrich) (100 ng/mL) were collected for RNA extraction using an RNeasy mini kit (Qiagen) and the samples were processed with the RNA sequencing of BGISEQ-50 platform by the BGI Company as previous published [93]. Briefly, the sequencing quality was determined using FastQC. The abundance transcripts were quantified by Kallisto and the transcripts were converted to genes by Tximport r package. The read count data were determined using Differential gene expression (DEGs) using edgeR. The fold-change (FC) 1 and -1 were used as the cutoff to choose DEGs. The up- and downregulated genes were analyzed in a Biological process (GO Term) using Enrichr. The gene lists associated with glycolysis, mitochondrial oxidative phosphorylation, and ATP synthesis were selected based on the GO Term of DEGs and the heat maps were generated by pheatmap r package.

### 4.6. Macrophage Phagocytosis and Bactericidal Activity

Macrophages phagocytosis and bactericidal activity were adapted from the previous protocols [94,95]. For phagocytosis, macrophages at 1 × 10^5^ cells were incubated with WGP (Biothera) 500 µg/mL with or without LPS (Escherichia coli 026: B6) (Sigma-Aldrich) at 100 ng/mL or DMEM control for 16 h before the 1 h incubation with zymosan conjugated with 40 kDa fluorescein isothiocyanate dextran (FITC-dextran) (Sigma-Aldrich) at 200 ug/mL (opsonized with 2.5% mouse serum in DMEM for 30 min before use) in 37 °C in 5% CO_2_. Then, cells were fixed by 1% paraformaldehyde in the PBS buffer containing 2% fetal bovine serum (FBS) for 15 min and cells were detached by a cell scraper. The number of phagocytosis cells was measured with a FACS LSRII (BD Biosciences, San Jose, CA, USA) flow cytometer. Data was analyzed using FlowJo software (Tree star, Inc., Asland, Oregon, USA) and the mean fluorescent intensity of FITC-positive cells were the representative of the macrophages with active phagocytic activity. For bactericidal activity, the macrophages at 1 × 10^5^ cells after 16 h treatment were incubated with the viable Escherichia coli (*E. coli*) (ATCC 8739) (Fisher Scientific), after the opsonization with 2.5% mouse serum in DMEM for 30 min before use, at the ratios of 100 bacteria per macrophage in 37 °C in 5% CO_2_ for 2 h. After that, the lysis buffer was applied and the samples were incubated with 3-[4,5-dimethylthiazol-2-yl]-2,5-diphenyltetrazolium bromide (MTT) for 1 h. The purple color of formazan was solubilized by Dimethyl sulfoxide (DMSO) and measured by a spectrometry (microplate reader) (Thermo Scientific) using the absorption wavelength at 570 nm. Notably, only the viable bacteria from lysed macrophages are able to react with MTT and convert color of MTT into the insoluble purple formazan, which is directly related with the viable bacteria that were not killed by macrophages.

### 4.7. Extracellular Flux Analysis

The energy metabolism profiles of macrophages were estimated by glycolysis and mitochondrial oxidative phosphorylation on the basis of the extracellular acidification rate (ECAR) and the oxygen consumption rate (OCR), respectively, by Seahorse XF Analyzers (Agilent, Santa Clara, CA, USA) as previously published [8,30,96]. Macrophages in different treatment groups were dispersed into monolayers for the measurement. For mitochondrial stress test, assays were performed in Seahorse XF DMEM medium (Agilent Technologies) supplemented with 2 mM Seahorse XF L glutamine, 1 mM pyruvate, and 10 mM XF Glucose. Then, cells were sequentially incubated by Oligomycin, Carbonyl cyanide 4-(trifluoromethoxy) phenylhydrazone (FCCP) and Rotenone/antimycin A (final concentration of 1.5 µM, 1 µM and 0.5 µM, respectively). In parallel, glycolysis stress tests were performed in Seahorse XF DMEM medium supplemented with 2 mM Seahorse XF L glutamine. Accordingly, several agents, including glucose, Oligomycin, and 2-Deoxy-d-glucose (2-DG) were sequentially injected to achieve final concentration of 10 mM, 1 µM and 50 mM during the procedure. In addition, the production of Adenosine triphosphate (ATP) from glycolysis and mitochondria was measured by Real-time ATP rate assay kit (Agilent). In brief, the treated macrophages were cultured in DMEM medium (Agilent Technologies) supplemented with 2 mM Seahorse XF L glutamine, 1 mM pyruvate and 10 mM XF Glucose. After that, Oligomycin and Rotenone/AntimycinA (as above) were added. The results in Seahorse analysis were normalized by applying the total protein abundance in the Wave program to measure all parameters.

### 4.8. Statistical Analysis

Statistical differences among groups were examined using the unpaired Student’s t-test or one-way analysis of variance (ANOVA) with Tukey’s comparison test for the analysis of experiments with two groups or more than two groups, respectively, and are presented as the mean ± standard error (SE). The time-point experiments were analyzed by the repeated measures ANOVA. All statistical analyses were performed with SPSS 11.5 software (SPSS, Chicago, IL, USA) and Graph Pad Prism version 7.0 software (La Jolla, CA, USA). A *p*-value of < 0.05 was considered statistically significant.

## 5. Conclusions

Candida-induced dysbiosis in bilateral nephrectomy mice enhanced gut translocation of (1→3)-β-D-glucan (a major component of fungal cell wall) that accelerated macrophage energy status and enhanced LPS responses. The importance of gut fungi and gut leakage in acute kidney injury and the utilization of BG in macrophage metabolic reprogramming are interesting for the clinical translation.

## Figures and Tables

**Figure 1 ijms-22-05031-f001:**
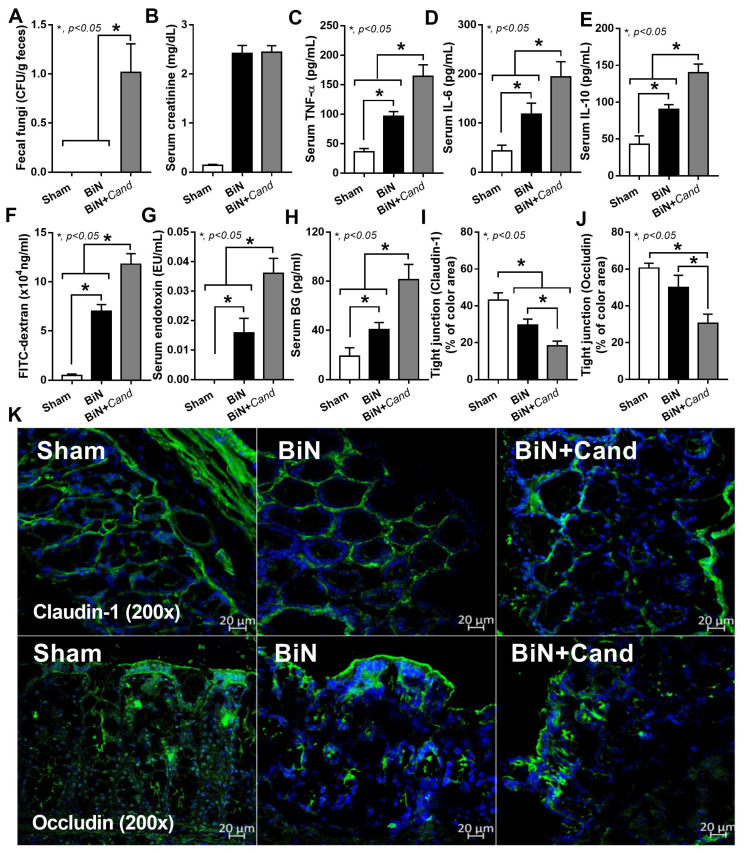
Characteristics of mice with sham, bilateral nephrectomy (BiN) and BiN with Candida administration (BiN+Cand) at 48 h after surgery, as indicated by fungal abundance in feces (culture method) (**A**), serum creatinine (**B**), serum cytokines (**C**–**E**), gut leakage; FITC-dextran, endotoxemia and serum (1→3)-β-D-glucan (BG) and the abundance of tight junction molecules (Claudin-1 and Occludin) as evaluated by immunofluorescent staining and scored in percentage of the detected-area with the representative pictures (**F**–**K**) are demonstrated (n = 7–9/group). Notably, data of sham with Candida administration (Sham+Cand) are not demonstrated due to the similarity to the sham group without Candida.

**Figure 2 ijms-22-05031-f002:**
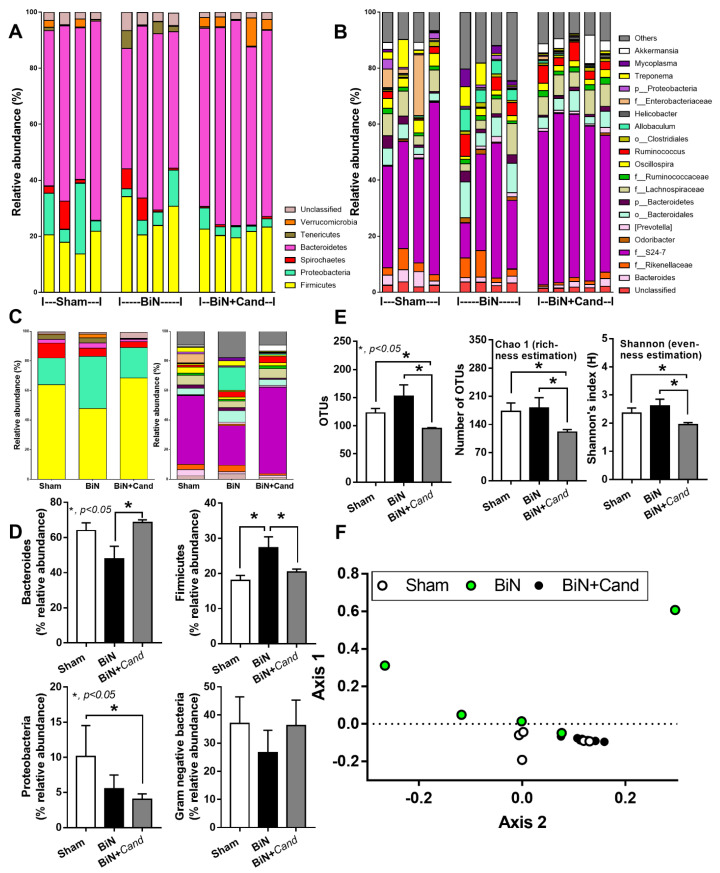
Fecal microbiome analysis from mice with sham, bilateral nephrectomy (BiN) and BiN with Candida administration (BiN+Cand) at 48 h after surgery as indicated by relative abundance of bacterial diversity at phylum level, genus level, and the average of both levels (**A**–**C**), the heterogeneity of fecal bacteria by operational taxonomic units (OTUs) and alpha-diversity indices (Chao and Shannon) (**D**), the comparison of bacterial diversity in phylum level in graphs (**E**) and the non-metric multidimensional scaling (NMDS) based on Thetayc beta diversity matrices at the genus level (**F**) are demonstrated (n = 5–6/group). Notably, data of sham with Candida administration (Sham+Cand) are not demonstrated due to the similarity to the sham group without Candida.

**Figure 3 ijms-22-05031-f003:**
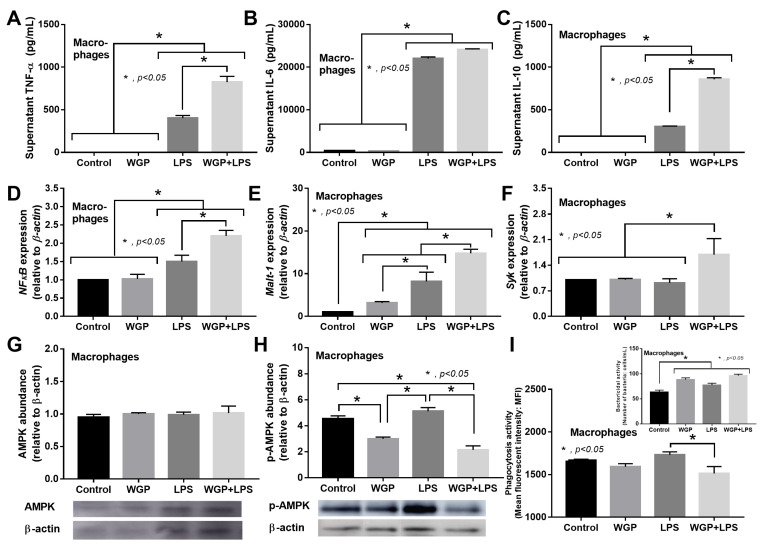
Characteristics of macrophage responses after 6 h incubation of media control (control), Whole Glucan Particle (WPG) with or without lipopolysaccharide (LPS) as indicated by supernatant cytokines (**A**–**C**), expression of the downstream signaling molecules (*NFκB*, *Syk* and *Malt**-1*) (**D**–**F**), protein abundance of AMPK, p-AMPK and a ratio between AMPK/p-AMPK (a sensor of cell energy status) with the representative figures of Western blot analysis (**G**,**H**), phagocytosis and bactericidal activity (**I**) are demonstrated (independent triplicate experiments were performed).

**Figure 4 ijms-22-05031-f004:**
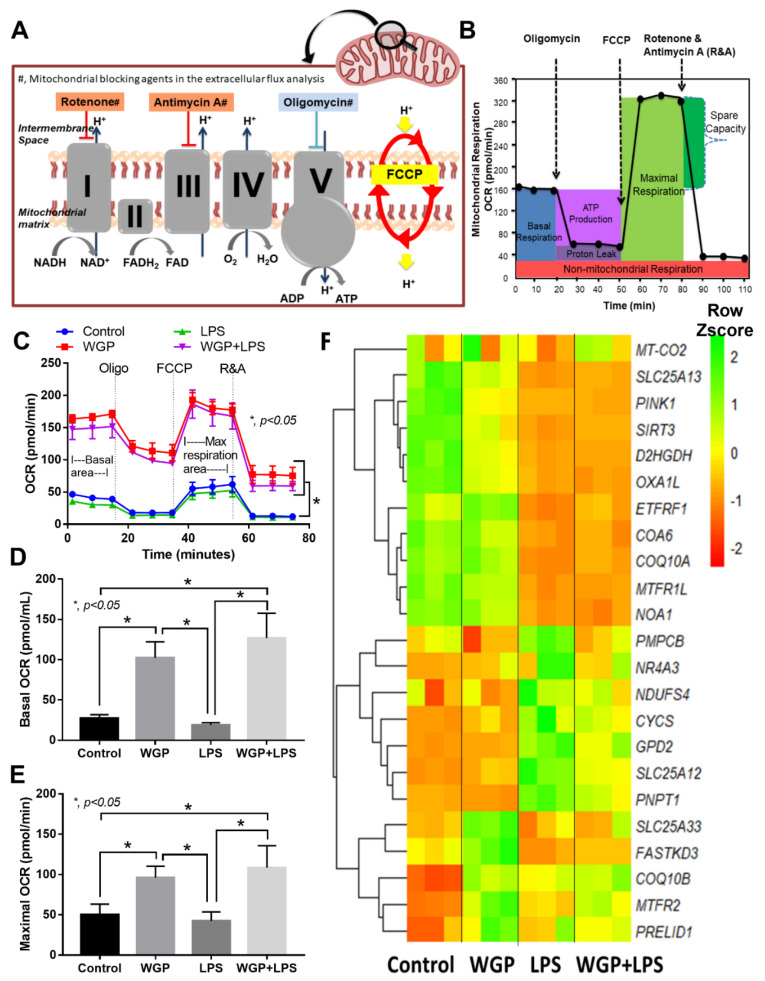
Graphic pictures indicate the action of mitochondrial inhibitory agents, including Rotenone and Antimycin A (the blockage of protein complex I and complex III, respectively) and oligomycin (a blockage of protein complex IV), and the activity of Carbonyl cyanide 4-(trifluoromethoxy)phenylhydrazone (FCCP), a mitochondrial augmented-agent through mitochondrial uncoupling (a process that transfers proton back to mitochondrial matrix to increase mitochondrial ATP synthesis) (**A**) and area of the graph from extracellular flux analysis, including several parameters of mitochondrial respiration based on oxygen consumption rate (OCR) (**B**). Characteristics of macrophage responses after 3 h incubation of media control (control), Whole Glucan Particle (WGP) with or without lipopolysaccharide (LPS) (the representative cell wall molecules of fungi and bacteria in gut) as evaluated by OCR mitochondrial respiration (**C**) with the better visualization on graph presentation of basal respiration and maximal respiration (**D**,**E**) and the heat-map of mitochondria-associated genes from RNA sequencing analysis (**F**) are demonstrated (independent triplicate experiments were performed). Notably, other OCR parameters, including proton leak, ATP production, and non-mitochondrial respiration are not shown due to the similar patterns as basal respiration and maximal respiration.

**Figure 5 ijms-22-05031-f005:**
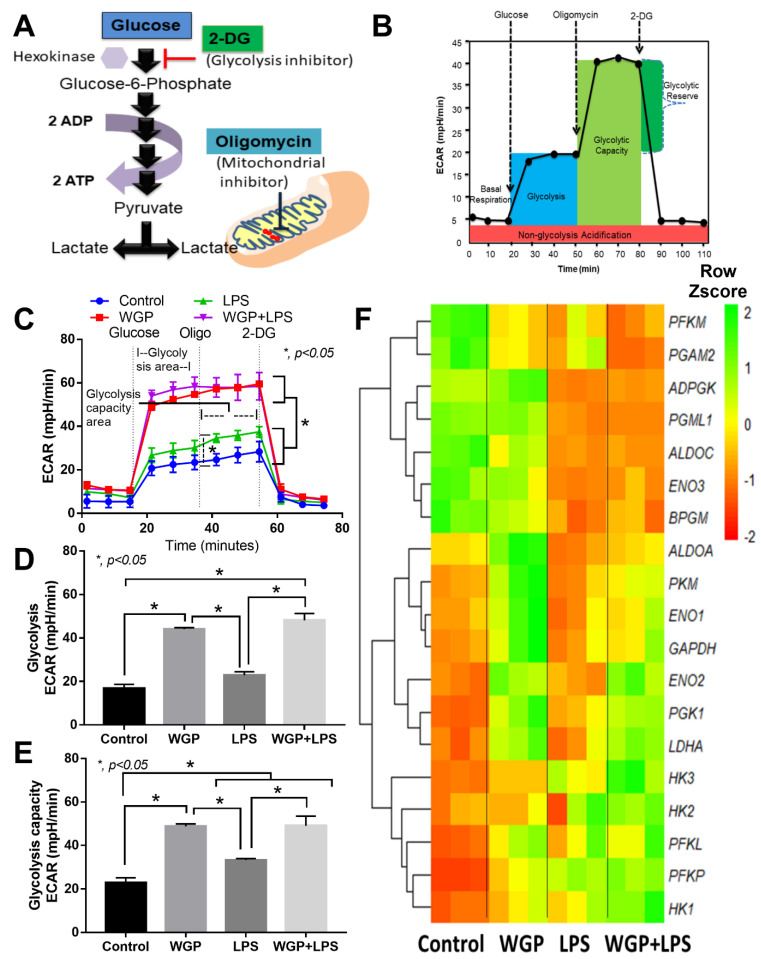
Graphic pictures indicate the action of a glycolysis inhibitory agents, 2-Deoxy-d-glucose (2-DG) on hexokinase (**A**) and area of the graph from extracellular flux analysis, including several parameters of glycolysis activity based on extracellular acidification rate (ECAR) (**B**). Characteristics of macrophage responses after 3 h incubation of media control (control), Whole Glucan Particle (WPG) with or without lipopolysaccharide (LPS) (the representative cell wall molecules of fungi and bacteria in gut) as evaluated by ECAR glycolysis activity (**C**) with the better visualization on graph presentation of glycolysis and glycolysis capacity (**D**,**E**) and the heat-map of glycolysis-associated genes from RNA sequencing analysis (**F**) are demonstrated (independent triplicate experiments were performed).

**Figure 6 ijms-22-05031-f006:**
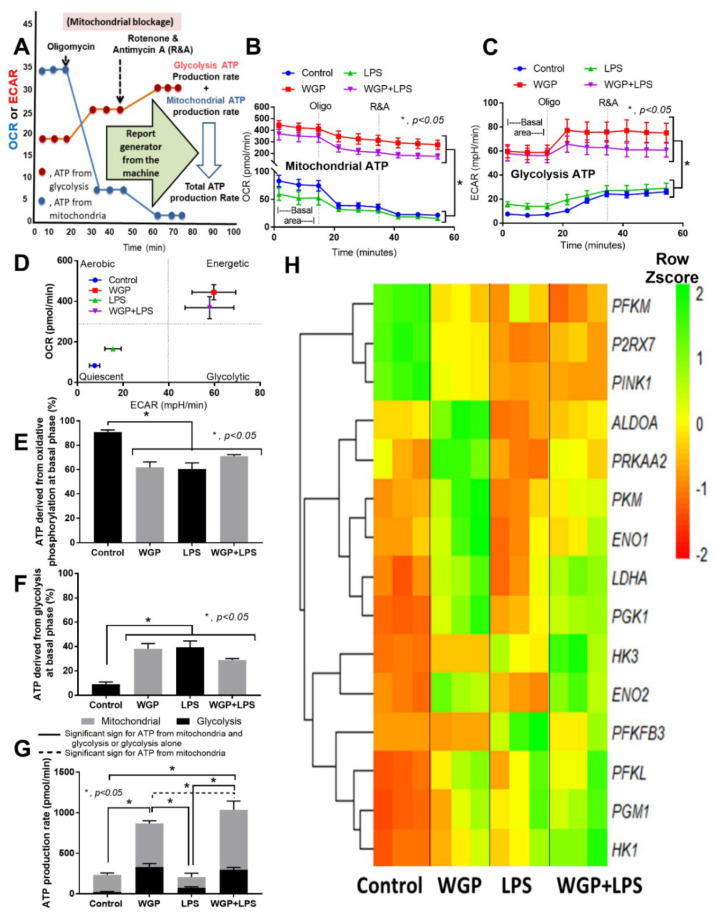
A graphic picture indicates the ATP analysis (**A**) that separates ATP from mitochondria and glycolysis through oxygen consumption rate (OCR) and extracellular acidification rate (ECAR), respectively, by the extracellular flux analysis machine using mitochondrial inhibitory agents, Oligomycin and Rotenone with Antimycin A (R&A) is demonstrated. The measurement is based on the basic knowledge that a blockage of mitochondrial ATP accelerates ATP production from the glycolysis pathway. Additionally, characteristics of macrophage responses after 3 h incubation of media control (control), Whole Glucan Particle (WPG) with or without lipopolysaccharide (LPS) (the representative cell wall molecules of fungi and bacteria in the gut) as evaluated by ATP production from either mitochondria (by OCR) or glycolysis (by ECAR) (**B**,**C**), the cell energy phenotype profile (by OCR versus ECAR) (**D**), percentage of ATP that derived from mitochondria (oxidative phosphorylation) (**E**) and from glycolysis (**F**), ATP production rate from mitochondria and glycolysis (**G**) and the heat-map of genes for ATP production from RNA sequencing analysis (**H**) are demonstrated (independent triplicate experiments were performed).

**Figure 7 ijms-22-05031-f007:**
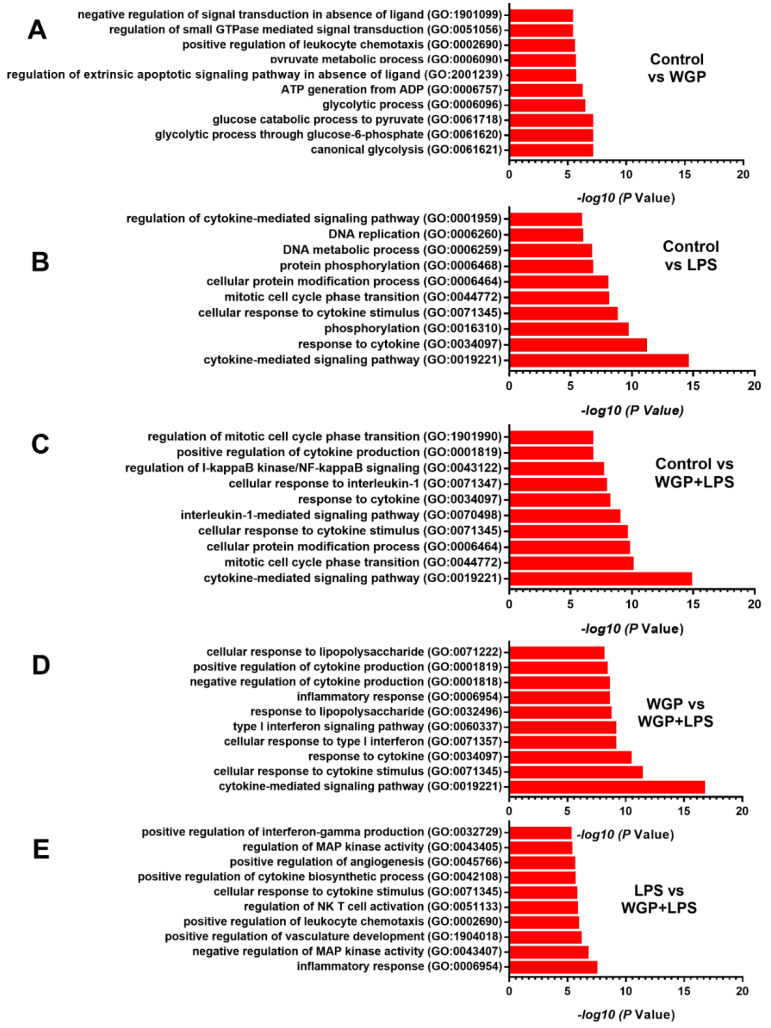
The biological process gene ontology terms (Go term) enrichment analysis of differential gene expression (DEGs) in macrophages with various conditions. The up- and downregulated genes (−1 to 1-fold change cut-off value) were identified using ontology enrichment analysis and the top ten significantly Go term with their –log10 p value in comparison between macrophages with control condition (Control) versus the activation by Whole Glucan Particle (WPG) with or without lipopolysaccharide (LPS) (the representative cell wall molecules of fungi and bacteria in gut) (**A**–**C**) and between WGP plus LPS (WGP+LPS) versus the activation by WGP or LPS alone (**D**,**E**) are demonstrated.

**Figure 8 ijms-22-05031-f008:**
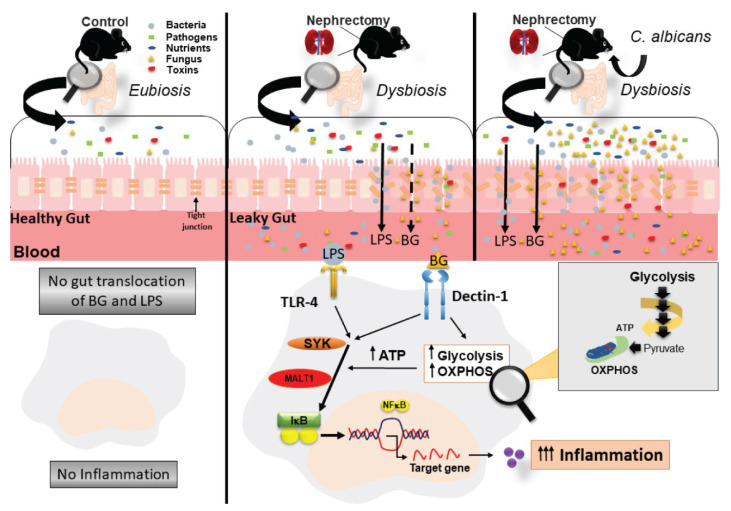
The proposed hypothesis demonstrates uremia-induced fecal dysbiosis that is more severe in bilateral nephrectomy (BiN) mice with Candida (BiN-Candida) than BiN alone (non-Candida BiN). There is a lesser translocation of (1→3)-β-D-glucan (BG) in non-Candida BiN mice than BiN-Candida (because of the increased BG level in feces of BiN-Candida mice), with a similar LPS translocation in both models. Then, LPS+BG (LPS with BG) activates TLR-4 and Dectin-1, respectively, and upregulates NFκB, a transcriptional factor for cytokine production, through Syk and Malt-1. In parallel, BG also enhances the important enzymes in glycolysis pathway (in cell cytoplasm) and in OXPHOS (oxidative phosphorylation) of mitochondria that increase cell energy status (ATP production) and enhance LPS responses.

**Table 1 ijms-22-05031-t001:** List of Primers in the study are demonstrated.

Primers	Forward	Reverse
*NF* *κ* *B*	5’-CTTCCTCAGCCATGGTACCTCT-3’	5’-CAAGTCTTCATCACATCAAACTG-3’
*Malt* *-* *1*	5’-CACACTGAGGTTCTTCCGCT-3’	5’-TGATGCATTCGGGGGCGTAC-3’
*Syk*	5’-CTACTACAAGGCCCAGACCC-3’	5’-TGATGCATTCGGGGGCGTAC-3’

*NF**κB*, Nuclear factor-κB; *Malt**-1*, Mucosa-associated lymphoid tissue lymphoma translocation protein 1; *Syk*, Spleen tyrosine kinase.

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
