# Peer review of "Candida Administration in Bilateral Nephrectomy Mice Elevates Serum (1→3)-β-D-glucan That Enhances Systemic Inflammation Through Energy Augmentation in Macrophages"

_ijms, 2021, doi:10.3390/ijms22095031_

Round 1

Reviewer 1 Report

This manuscript describes the results of essentially 2 different experiments. Broadly defined, the manuscript demonstrates the effect of colonization with Candida albicans on nephrectomized mice, in terms of induction of inflammation and gut permeability. There are marked differences based on Candida colonization status in these animals, with the presence of C. albicans associated with increased systemic inflammation and gut permeability in nephrectomized animals. They also present interesting data on differences in GI microbiota in these animals. The subsequent experiments are conducted in vitro with bone-marrow derived mouse macrophages, and interrogate the impact of whole glucan particles and LPS alone and in combination on a variety of macrophage functions. An in depth analysis of the metabolic/cell energy pathways in these macrophages is also provided. The data presented are extensive and the experiments are generally sound. However, the manuscript suffers from a few elements that limit the potential impact of the work. The following comments are offered for the authors' consideration: 1. The experiments in the mouse nephrectomy model are novel and informative. However, the extent to which the subsequent experiments using ex vivo macrophages relate to the central theme is very questionable. Combining all of these experiments into a single manuscript confuses the central message. For example, the title "Enhanced macrophage responses to endotoxin in Candida-administered bilateral nephrectomy mice through the cell energy augmentation by...glucan" is not truly supported by the data. The macrophage responses and the cell energy effects of glucan are demonstrated, but these experiments were not conducted in Candida-administered nephrectomized mice. It is plausible that the effects demonstrated in the animal model are mediated by the metabolic effects of glucan, but this was not tested directly in the experiments presented. The authors might consider presenting these experiments in 2 distinct manuscripts in order to avoid conflating the results in ways that may not be directly supported by the data. 2. Overall the manuscript suffers from atypical grammar and English usage to the extent that the overarching themes, hypotheses, and conclusions become difficult to discern. Careful and extensive editing by a native English speaker is strongly recommended.

Reviewer 2 Report

The goal of the present study was to define metabolic changes of macrophages in Candida-worsen acute kidney injury in mouse model. Results from fundamental molecular methods used were interpreted mostly clearly and proved hypothesis of the article. From my point of view, the manuscript meets requirements on the novelty, although the fact that the obtained data about enhanced cell energy status and ATP obtained mainly from glycolysis in Mfs are not very surprising. Moreover, the manuscript has the following drawbacks, and can be published only with some improvements.

Major:

  • There is lot of data obtained from Seahorse, please declare which Seahorse kit did you use and more importantly: I miss Seahorse data normalisation, did you apply it (for example normalisation for cell count by CyQUANT®)? If not, please add.
  • The topic of M1 and M2 polarisation is by some author considered as outdated, however you are applying this functional polarisation in your manuscript. Nevertheless, there are almost no significant data supporting this dividing (e.g. there is no evidence in iNOS activation and/or Arg-1 downregulation), so I would recommend their exclusion.

Minor:

  • Please extend the general information about the problematics in the abstract - for non-familiar reader is difficult to understand the topic from 1 brief sentence and lot of result information.
  • Please discuss, if is p-AMPK stated in the text as the sensor of cell energy status or even marker of mitochondrial respiration as mentioned in previous literature; please state, why is p-AMPK compared to B-actin and not to total AMPK. Please correct the statement (r. 155 and 156): “Because LPS alone or in combination with WGP reduces p-AMPK (a sensor of cell energy status) in macrophages (fig 3G), the enhanced cell energy status after LPS induction is possible.” – the LPS did not have stated effect.
  • Please make sure, that references to the figures 4 and 5 in results match to the correct figures, resp. their "sub-letters", it seems that there are some minor errors.
  • You stated in discussion that there is correlation between Mfs cell energy and cytokine production. Please prove this statement and discuss deeply the mechanism of interaction.

Round 2

Reviewer 2 Report

In my opinion, authors improved the article according to recommendations from first submission and explained misunderstandings and errors and I agree with publication in presented form.